# Intranasal Inoculation with Classical Swine Fever Virus Provided a More Consistent Experimental Disease Model Compared to Oral Inoculation

**DOI:** 10.3390/vetsci11020056

**Published:** 2024-01-28

**Authors:** Mette Sif Hansen, Jens Nielsen, Åse Uttenthal, Gitte Øland Jensen, Louise Lohse

**Affiliations:** National Veterinary Institute, Technical University of Denmark, Lindholm, DK-4771 Kalvehave, Denmark; jeni@aqua.dtu.dk (J.N.); lolo@ssi.dk (L.L.)

**Keywords:** pigs, classical swine fever virus, experimental model, Paderborn strain, intranasal vs. oral inoculation, immunomodulation, CRP, TNF-α, leukocyte sub-populations

## Abstract

**Simple Summary:**

Classical swine fever is a serious virus infection in pigs, and outbreaks of the disease result in major problems with regard to animal welfare, as well as to the economy of the farmers. In order to control the disease, a lot of research has been carried out for decades. For classical swine fever, it has been demonstrated that the severity of the disease is determined by several factors, including virus strain and host factors, such as the age and immunity of the pigs. However, many studies have provided divergent results, making interpretation and conclusions difficult, so the putative influence of the experimental setup on the outcome of the infection has attracted increasing attention. For classical swine fever experimental studies, infection of the pigs is usually performed by the application of the virus in the mouth or the nose. In our study, we examined a number of parameters, including clinical disease and indicators of disease, in pigs infected orally or intranasally. The demonstrated wide variation in disease outcomes after oral application of the virus indicates that this method is less suitable for comparative studies. These observations provide additional information on the mechanisms of classical swine fever infection that are important for combatting the disease.

**Abstract:**

The severity of disease resulting from classical swine fever virus (CSFV) infection is determined by several factors, including virus strain and host factors. The different outcomes of experimental studies in pigs with the same strain of CSFV emphasize the need to elucidate the influence of individual factors within experimental protocols. In this study, we investigated the outcome of disease after oral and intranasal inoculation with a moderately virulent CSFV strain in young pigs. To compare the two routes of inoculation, various infection parameters were examined during a period of two weeks. While all intranasally inoculated pigs (*n* = 5) were directly infected, this was only the case for two out of five pigs after oral inoculation. In addition, the intranasally inoculated pigs developed a more pronounced clinical disease and pathological lesions, as well as markedly more change in hematological and immunological parameters than the orally inoculated pigs. The wide variation among the orally inoculated pigs implied that statistical evaluation was markedly impaired, leaving this route of application less suitable for comparative studies on classical swine fever. Furthermore, our study provides additional details about the immunomodulatory effects of CSFV on the kinetics of CRP, TNF-α, and leukocyte sub-populations in pigs after infection with the CSFV strain Paderborn.

## 1. Introduction

Infection of pigs with classical swine fever virus (CSFV), an RNA virus belonging to the genus Pestivirus within the family Flaviviridae [1], can cause severe disease. The pathology of CSF infections is dominated by hemorrhagic syndrome and immunosuppression [2,3]. The former include petechial bleeding of the skin and mucosae, as well as spleen infarction. Immunosuppression is characterized by lymphocyte depletion and depressed T cell activity, as well as regressive changes in lymphoid organs [4,5,6,7,8,9]. Due to central nervous system infection, pigs may also show progressive depression and uncoordinated movements [10,11].

It is, however, generally accepted that the clinical and pathological outcome of CSFV infections depends on several factors, including the virulence of the virus strain, and factors relating to the host, e.g., age, breed, genotype, and immune and sanitary status [4,12,13,14,15,16]. In addition, the severity of the outcome of experimental CSFV infections may vary not only between different breeds but also between individual pigs of the same age and breed [13,17,18,19]. The existence of strains with varying virulence from avirulent to high virulence is well established. However, the virulence of CSFV is a controversial topic of discussion since the characterization of virulence has been attempted in various ways and the influence of factors not associated with the virus strain per se has only been sparsely studied in harmonized experimental settings [4,13,20].

Under natural conditions, pigs are usually infected with CSFV orally and/or intranasally [21], and a number of studies indicate that the route of inoculation may influence the outcome of disease in experimentally infected pigs. For example, the CSFV strain Eystrup has been characterized as highly virulent in 30–35 kg body weight specific pathogen-free (SPF) pigs (inoculated by oral or intranasal routes or a combination) [20,22]. However, in our laboratory, oral inoculation using equivalent doses of the same Eystrup virus stock resulted in severe disease in 5-week-old conventional pigs (Uttenthal and Nielsen, unpublished results) but only moderate or mild disease in 6- to 11-week-old conventional pigs [23] and pigs of high sanitary status [4]. In another study, using different inoculation routes [18], German pigs infected oronasally with the 10^3^ TCID_50_ CSFV strain Paderborn developed acute infection with high mortality (80%), whereas French and Danish pigs infected intranasally (10^2.54^ TCID_50_) or orally (10^4.4^ TCID_50_), respectively, displayed a milder but more chronic infection and a later onset of disease. Also, interestingly, the French pigs developed clinical signs characterized by effects on the central nervous system with ataxia, whereas the Danish pigs showed skin hemorrhages, apathy, and diarrhea. Thus, it seems possible that the different routes of inoculation may have played a role in producing the observed differences in the severity of disease between the various studies on the CSFV strain Eystrup [4,20,24] and the CSFV strain Paderborn [18]. Intramuscular injection may also be used for CSFV application; however, this method is mainly used in CSF vaccine studies [25,26].

In the present study, we elucidated the outcome of disease after oral and intranasal inoculation of Danish SPF pigs with the CSFV strain Paderborn in a controlled experimental setting. To compare the two individual infection models, we examined a range of clinical, pathological, serological, virological, hematological, and immunological parameters that are characteristic of swine fever pathogenicity [3,5,19,20].

## 2. Materials and Methods

### 2.1. Animals

Ten 5-week-old cross-bred SPF pigs from a commercial swine herd were transferred to the animal biosafety level 3 isolation facilities at the National Veterinary Institute, Lindholm. The pigs were randomly selected from a uniform pool of pigs from eight litters that had been mingled after weaning at 4½ weeks of age.

### 2.2. Experimental Setting

The pigs were randomly divided into two groups, each with five pigs, which were housed in two separate rooms. At 6 weeks of age (post-inoculation day (PID) 0), one group of pigs (ORAL; Nos. 1–5; three females, two males) was inoculated orally by depositing 10^5^ TCID_50_ of the moderately virulent CSFV strain Paderborn [18] in 4 mL Eagle’s medium onto the surface of the tonsils, the primary target organ for CSFV, as previously described [4]. The other group (NASAL; Nos. 6–10; two females, three males) was inoculated intranasally with an equivalent dose of the same pool of virus: 2 mL in each nostril. The inoculum, which consisted of serum from a pig passage of CSFV-277, was then back-titrated on PK-15 cells and found to have a titer of 10^4.4^ TCID_50_ per 4 mL (1 pig dose). Blood samples were collected from the anterior vena cava in vacutainers from all pigs on PID 0, 3, 7, 10, and 14 for virological, hematological, serological, and immunological examination. EDTA-stabilized blood was used for total white blood cell (WBC) counts, leukocyte differentiation, and phenotyping by flow cytometry. Serum was used for measurements of C-reactive protein (CRP) and TNF-α and the detection of CSFV RNA for virus isolation and antibody analysis. On PID 3, 7, 10, and 14, tonsil scrapings were obtained with a blunt stainless steel scraper. To avoid artificial damage to the tonsils before inoculation and thereby potentially affecting the outcome of the inoculation, tonsil scraping was not carried out before the inoculation. The experiment ended on PID 14 when all pigs were sedated and then euthanized by intravenous injection of Pentobarbital. The animal experiment was performed according to the experimental protocol approved by the Danish Animal Experiments Inspectorate, license No. 2003/561-742. 

### 2.3. Clinical Examination and Pathology

Clinical observations of individual pigs were recorded on a daily basis. On PID 3, a microchip transponder was inserted subcutaneously behind the left ear of each pig for daily measurements of body temperatures from PID 0, as previously described [27]. Rectal body temperatures were measured on PID 1 (NASAL group), 3 (ORAL group), and in both groups on PID 8, 11, and 14; i.e., early in the experiment for assessment of the consistency of the transponder temperature measurements, and later on, when infection and clinical disease progressed, for a close monitoring of the individual pigs for animal welfare considerations. The correlation between transponder temperature (tempT) and rectal temperature (tempR) can be expressed as tempT = 4.0848 + 0.8755 × tempR [27]. Fever was defined as rectal temperatures (tempR) equal to or above 40.0 °C, equivalent to tempT of 39.1 °C. To obtain a semi-quantitative measurement for comparison of clinical disease between the two groups, all pigs were scored daily using a slightly modified version of the clinical scoring (CS) system developed by Mittelholzer et al. [20]. Thus, 9 clinical parameters (liveliness; body tension; body shape; breathing; walking; skin; eyes and conjunctiva; appetite; defecation) were evaluated and scored on a 0-to-3-point scale, where 0 reflects the absence of CSF symptoms and 3 represents a severe level of a CSF symptom.

Following euthanasia, the pigs were weighed, and a necropsy was performed for the characterization of gross pathology. A pathological score was estimated for each pig according to a scoring system developed by Floegel-Niesmann et al. [13]. Nine tissue compartments (skin; subcutis and serosa; tonsil; spleen; kidney; lymph nodes; ileum and rectum; brain; respiratory system) were examined for lesions associated with CSF and scored on a 0 (no lesion) to 3 (severe CSF lesion) scale. In addition, the thymus (cervical and thoracic parts) was excised and weighed, and the thymus (g)/body weight (kg) ratio (T/BW) was estimated as a measure of thymus atrophy [28], i.e., normal thymus (T/BW ≥ 2); mild atrophy (T/BW = 2–1.5); moderate atrophy (T/BW = 1.5–0.5); or severe atrophy (T/BW ≤ 0.5).

### 2.4. Virus Detection

Tonsil scrapings and serum samples were analyzed for the presence of CSFV by virus titration on PK-15 cells [18]. The serum samples were also examined for the presence of CSFV-RNA by quantitative real-time RT-PCR [23], and Ct values equal to or above 40 were regarded as negative.

### 2.5. Serological Examination

Serum samples were assessed for antibodies to CSFV using a blocking ELISA [29]. The levels of CRP and TNF-α in serum were determined with an in-house indirect ELISA [30] and a solid-phase sandwich ELISA (KSC3011, Invitrogen, Camarillo, CA 93012, USA), respectively.

### 2.6. Hematological and Immunological Parameters

#### 2.6.1. WBC

Determination of the number of WBCs was performed on EDTA-stabilized blood samples using a semi-automated animal blood cell counter (Vet abcTM, ABX, Montpellier, France). All samples were counted twice, and the mean value was calculated.

#### 2.6.2. Flow Cytometry Analyses

For the phenotyping of leukocytes in peripheral blood, flow cytometry methods were employed, as previously described [4]. Single labeling was used to identify lymphocytes, monocytes, granulocytes, and B cells, and triple labeling was used to identify T cell sub-populations, defined as CD3^+^CD4^+^CD8^−^ naïve T helper (Th) cells, CD3^+^CD4^+^CD8^+^ memory/activated Th (Tm) cells, CD3^+^CD4^−^CD8^+^ cytotoxic T (Tc) cells, CD3^+^CD4^−^CD8^low/−^ γδ T cells, and CD3^−^CD4^−^CD8^+^ natural killer (NK) cells [5].

### 2.7. Statistical Analysis

Data analysis was performed by Student’s *t*-test using GraphPad In Stat version 3.00 (GraphPad Software, San Diego, CA, USA). An unpaired *t*-test was used to compare the means of the ORAL and the NASAL group. A paired *t*-test was used to compare changes over time referring to PID 0 within the two groups. Differences were considered significant at *p* < 0.05. The error bars on the graphs represent the SD.

## 3. Results

### 3.1. Clinical Observations

From PID 1 or 2, all orally inoculated pigs shed semi-liquid feces for 4 to 5 days. Pig Nos. 1 and 2 became slightly depressed with forced respiration, seizures, and/or diarrhea from PID 7 and 10, respectively. Amongst the intranasally inoculated pigs, semi-liquid feces (Nos. 7, 9) and slight depression (Nos. 8, 10) were observed during the first week. For the remaining period of this study, all pigs in this group were slightly depressed, had semi-liquid feces, and reduced appetite. On PID 8-10 until the end of the experiment, the condition of pig Nos. 7, 8, and 10 gradually deteriorated as they became increasingly lethargic and anorexic and developed watery diarrhea, stiff gait, forced respiration, and reddening of the conjunctiva. The clinical scores are shown in Figure 1. Compared to the ORAL group, pigs in the NASAL group had significantly increased mean CS on PID 7 onwards (*p* < 0.05–0.001). In addition, a total of 36 cumulative days with fever were recorded in the NASAL group in contrast to only 13 days with fever in the ORAL group (Figure 1). 

### 3.2. Necropsy Findings

At necropsy, the pigs mainly presented with only a few lesions typical of CSFV infection (Figure 2). Macroscopically, the thymuses of all but pig No. 2 were considered to be atrophic, and this was generally most pronounced for the pigs in the NASAL group. The T/BW ratio, which defined the degree of thymus atrophy in individual pigs as mild (No. 4), moderate (Nos. 1–3, 5 and 6, 7, 10), or severe (Nos. 8, 9). Other findings were circulatory disturbances, such as petechia, in the kidneys in one orally inoculated (No. 2) and three intranasally inoculated pigs (Nos. 8–10), and bleeding in the mandibular or intestinal lymph nodes (Nos. 2–4, 10). Hyperemia was observed in the lepto-meninges of five pigs (Nos. 2, 3, 6–8), in the bladder of one pig (No. 6), and in the spleen of pig Nos. 6 and 10, the latter also had infarcts in the spleen characteristic of CSF. The tonsils were hyperemic, had bluish discoloration, pustules, and/or were atrophic in five pigs (Nos. 1, 2, 8, 9, 10). Seven pigs (Nos. 2–4, 6–8, 10) displayed signs of diarrhea with greenish and watery intestinal content. Other lesions were bronchopneumonia in pig Nos. 2, 7, and 8, and ascites in pig Nos. 8 and 10. In general, lesions were more frequent and pronounced in the intranasally inoculated pigs compared to the orally inoculated pigs. This was supported by the trend toward a higher mean pathological score (mean score 5.4) in the NASAL group compared to the ORAL group (mean score 3.2); however, the difference was non-significant (Figure 3).

### 3.3. Virus Detection and Serology

The measurements of CSFV and CSFV-RNA in tonsillar scrapings and serum samples are presented in Table 1. In the ORAL group, CSFV and/or CSFV-RNA could first be detected in serum on PID 3, 7, and 14 in pig Nos. 2, 1, and 5, respectively. Pig Nos. 3 and 4 remained PCR- and virus-negative during the experiment. All the intranasally inoculated pigs were virus-positive in serum on PID 7 and throughout the rest of this study period. The viral load in serum was higher in the NASAL than in the ORAL group but comparable within groups. Viremia and virus isolation from the tonsils were markedly more frequent and consistent in the NASAL than in the ORAL group.

All serum samples were negative for antibodies against CSFV throughout the experiment.

### 3.4. CRP

The CRP levels in serum were low (<5.86 µg/mL) on PID 0 and 3 for all pigs except No. 5, which had a slightly higher concentration of 14.36 µg/mL on PID 0. The mean CRP levels peaked on PID 7 in both groups with values of 4 (ORAL) and 12 (NASAL) times the pre-inoculation levels, followed by a decrease on PID 10 and a slight increase on PID 14 in both groups (Figure 4). The highest concentrations of CRP for individual pigs ranged between 26 and 54 µg/mL for the ORAL group and 58 and 62 µg/mL for the NASAL group on PID 7, 10, and 14. Compared to the levels in the ORAL group, the intranasally infected pigs had increased mean levels of CRP (*p* < 0.05) on PID 7 and 10.

### 3.5. TNF-α

The TNF-α pre-inoculation values were below 155 pg/mL for individual pigs in both groups (Figure 4 and Appendix A). On PID 7, mean TNF-α levels had increased in both groups to about 1.5–2 times that of the pre-inoculation values, but with marked individual differences ranging from 114 to 339 pg/mL in the ORAL and 218 to 315 pg/mL in the NASAL group. The mean levels then dropped in both groups, but among the orally inoculated pigs, there was a second slight increase on PID 14, corresponding to an increase in CRP for this group. The recorded fluctuation in the mean values of TNF-α among the orally inoculated pigs with a second peak on PID 14 was caused by de novo infection of pig No. 5 and a second increase of TNF-α levels in pig No. 2, which experienced clinical deterioration (increased clinical score) on PID 13 and 14. However, the mean TNF-α levels did not differ significantly between the two groups at any time.

### 3.6. Total and Differential WBC Counts

In both groups, the mean total WBC counts gradually decreased over time from 23.4 to 25.0 × 10^6^/mL on PID 0 in the ORAL and NASAL groups, respectively (Figure 5), to levels significantly lower on PID 14 in the ORAL group (*p* < 0.0001) and on PID 7, 10, and 14 in the NASAL group (*p* < 0.01–*p* < 0.001). On PID 14, all virus-positive pigs had reduced levels of WBC to approximately 10 × 10^6^/mL. The kinetics of the change in total WBC counts were paralleled by decreased lymphocyte counts on PID 14 in the ORAL group (*p* < 0.01) and on PID 7, 10, and 14 in the NASAL group (*p* < 0.0001) compared to PID 0 (Figure 5). While granulocyte counts gradually decreased for both groups, significant reductions were found in the NASAL group on PID 7, 10, and 14 (*p* < 0.05–*p* < 0.01), but not in the ORAL group (Figure 5). However, the relative distribution between lymphocytes and granulocytes changed significantly in the NASAL group toward higher mean percentages of granulocytes on PID 7, 10, and 14 (*p* < 0.01, *p* < 0.001, *p* < 0.01) compared to PID 0. The monocyte counts had decreased in the ORAL group on PID 14 (*p* < 0.05) and in the NASAL group on PID 10 and 14 (*p* < 0.01 and *p* < 0.05) (Figure 5). Generally, the decreased cell counts were most pronounced in the NASAL group. However, the differences in the mean values for the total WBC, the lymphocytes, the monocytes, and the granulocytes were not statistically significant between the two groups at any time, reflecting the blurring effect of the pigs in the ORAL group that were not directly infected after inoculation.

### 3.7. Analysis of Blood Leukocyte Sub-populations

In both groups, the number of B cells gradually decreased over time. The decrease in mean B cell levels was only significant in the ORAL group on PID 14 (*p* < 0.01) and on PID 7, 10, and 14 (*p* < 0.001) in the NASAL group (Figure 6). On PID 7 and onwards, B cell numbers markedly dropped in all directly infected pigs in both groups and on PID 14 in the contact-infected pig No. 5 in the ORAL group (Appendix A). On PID 14, these eight pigs were almost depleted of B cells in peripheral blood, with B cell numbers between 0.013 and 0.084 × 10^6^/mL. Although the decline in B cells was markedly more rapid and lower in the NASAL compared to the ORAL group, the mean numbers of B cells did not differ significantly between the two groups at any time point, reflecting the high standard deviation in the ORAL group, where only two pigs were directly infected after inoculation. 

In general, the numbers of Th, Tc, Tm, and γδ T cells gradually decreased over time in both groups, with the most marked changes in the intranasally inoculated pigs and a minor delay in the orally infected pigs (Figure 6, Appendix A). A significant difference between the two groups for these five leukocyte sub-populations was detected only for the mean level of Th cells (*p* < 0.05) on PID 7. The number of Th cells showed significantly decreased levels over time in the NASAL group on PID 7 (*p* < 0.001), 10 (*p* < 0.01), and 14 (*p* < 0.001) and in the ORAL group on PID 14 (*p* < 0.05). Compared to PID 0, the number of Tc cells had decreased in the NASAL group on PID 7, 10, and 14 (*p* < 0.01). While the number of Tc cells decreased between 2.5 and 7 times over time in individual pigs, the percentage of Tc cells showed a temporary increase in five of the directly infected pigs, peaking on PID 7 (No. 1) and PID 10 (Nos. 2, 6, 7, and 8). Decreased numbers of NK cells were seen in both groups on PID 7, 10, and 14 (*p* < 0.01–*p* < 0.001) compared to PID 0. On PID 10, however, a temporary increase in the percentage of NK cells was seen compared to PID 7. The number of γδ T cells did not decrease significantly in the ORAL group over time. However, in the NASAL group, decreased levels of γδ T cells were recorded on PID 7, 10, and 14 (*p* < 0.05, *p* < 0.001, and *p* < 0.01, respectively). While the number of Tm cells had gradually decreased on PID 7, 10, and 14 (*p* < 0.01) in the NASAL group, a drop in Tm cells was not recorded until PID 14 (*p* < 0.01) in the ORAL group.

## 4. Discussion

In the present study, pigs inoculated orally with the CSFV strain Paderborn showed a marked individual variation for a range of examined parameters in contrast to the more uniform results observed for intranasally inoculated pigs. The lower infection rate after oral inoculation of the virus, with only two out of five pigs directly infected, in contrast to all pigs in the intranasally inoculated group, implied that comparative statistical evaluation was markedly impaired when pigs were orally inoculated. In addition, intranasally infected pigs appeared to develop more pronounced clinical disease and pathological lesions, as well as markedly more changes in the hematological and immunological parameters examined, than the orally inoculated pigs. The results did not indicate any correlation between the sex of the pig and the outcome of the disease.

Thus, the markedly more uniform infection rate among the intranasally compared to the orally inoculated pigs indicates that intranasal inoculation provides a more consistent virus inoculation route for comparative analyses of various CSF infection studies, as well as a high robustness of challenge experiments in vaccine testing. However, considering the limited number of pigs in each experimental group, the results should be verified by further studies, preferably using CSFV strains of different virulence. At the end of this study, the orally inoculated pig No. 5 became infected; however, this was likely due to virus excretion from the infected pigs (Nos. 1, 2) in the group. Summerfield et al. [5] showed that changes in leukocyte or lymphocyte sub-populations were observed up to 4 days prior to the onset of viremia in CSFV-infected pigs. In some infected pigs, lymphocyte depletion was also identified before clinical signs appeared. In our study, therefore, the markedly decreased numbers of B cells and Tc cells in pig Nos. 3 and 4 in the ORAL group at the end of the experiment may indicate that these two pigs were infected with virus transmitted from pig Nos. 1 and 2 that shed virus on PID 7-14. Such de novo infection by transmission from other infected pigs can be expected to result in a time lag in the course of infection, which further increases the lack of consistency between examined parameters. The apparently more pronounced alterations of several of the examined parameters, including higher CS values, higher body temperatures, and higher virus titers in serum, together with more severe pathological lesions in the intranasally infected pigs indicate that the inoculation route plays a role in the development of disease. In conclusion, our study supports the thinking that the harmonization of the route of virus application and the experimental design of CSF studies are of crucial importance to ensure the generation of reproducible and comparable data.

The reduced infection rate that we saw in the orally inoculated pigs is counterintuitive since oral inoculation is probably best at reflecting the main route of natural infection. This observation is also important when developing oral bait vaccines for the control of CSFV in wild boar populations [31]. In the present study, a moderately virulent CSFV strain only resulted in direct infection of 40% of orally inoculated pigs, indicating that the efficacy of CSFV bait vaccines might be challenged, which is supported by a previous study, showing that live oral CSFV vaccines do not always protect the pigs [32]. One reason for the reduced oral infection rate could be that oral inoculation stimulates saliva production, which clears the oral cavity of CSFV, whereas the intranasal inoculum persists in the nasal cavity for longer periods of time and thereby increases the tonsil exposure time. However, it remains to be investigated whether other factors associated with the individual pigs may play a role in the different disease outcomes of the two inoculation routes, e.g., the distribution and the number of viral receptors in the oral and nasal cavities and the influence of the innate mucosal immunity. 

In the present study, the clinical signs and gross lesions after oral and intranasal inoculation with the moderately virulent CSFV strain Paderborn resemble those previously observed within the first two weeks of infection in other studies [2,18]. The thymus atrophy that was observed in nine of ten pigs is consistent with previous findings in studies using the Alfort strain of CSFV [33,34] and with the highly virulent Koslov strain, where an almost total reduction in the size of the thymus was seen (J. Nielsen, unpublished results). Even in pigs with only mild disease after infection with the Eystrup strain, a size reduction of up to approximately 50% of the thymus was reported [4]. The moderate reduction in the thymus in pig No. 3 was associated with a marked reduction in B cells and Tc cells together with an increased level of TNF-α on PID 14. Altogether, these changes support the fact that this pig was infected by indirect transmission. Since thymus atrophy in juvenile pigs has been demonstrated after infection with CSFV strains of varying virulence, the atrophy of the thymus is suggested to be considered strongly indicative of CSFV infections in juvenile pigs.

Leukopenia, in particular lymphopenia, is a characteristic presentation during classical swine fever infections with CSFV strains of different virulence [6,7,19,34,35]. Summerfield et al. [5] described that pigs infected with highly and moderately virulent strains developed leukopenia, accompanied by lymphopenia involving leukocyte sub-populations in a disparate manner, with B cells, T helper cells, and cytotoxic T cells mostly affected. The latter changes were also observed in the infected pigs in our study with the moderately virulent Paderborn isolate, thus emphasizing the effect of CSFV infections on these cell populations. However, we also observed significantly decreased numbers of Tm, γδ T, and NK cells, thus emphasizing the effect of CSFV on these cell populations, too. The dramatic B cell depletion seen in the infected pigs in both experimental groups is in accordance with the depletion observed in severely and moderately [5,6], as well as mildly, affected [4] pigs and represents a characteristic event during CSFV infections per se. Taken together, our study supports previous observations that not only B lymphocyte depletion, but also T lymphocyte depletion is a general feature of CSF [5,6,19]. However, the temporarily increased percentages of Tc and NK cells observed in a number of infected pigs on PID 7 and 10, respectively, support considerations about the essential role of the cytotoxic cell populations in the antiviral host defense against CSFV [36].

A delayed or even absent humoral immune response is characteristic of CSFV infections [21]. The mechanism behind the lack of detectable CSFV antibodies in our pigs is likely to be associated with the demonstrated B cell depletion in a complex interplay with released cytokines, as previously suggested [37,38]. 

In pigs, haptoglobin and CRP are considered to be the major acute-phase proteins [30]. In our study, with a moderately virulent CSFV strain, the infected pigs had a considerably lower (3 times) CRP level than observed in pigs suffering from more severe CSF disease [39] but were comparable to the levels in pigs with milder symptoms [4]. This indicates that levels of CRP correlate with the severity of CSFV infections. 

Several studies have highlighted the important role of cytokines in the pathogenesis of CSFV infection [34,38,40,41]. Thus, TNF-α appears to be a major cytokine involved in the pathogenesis of the characteristic lymphocytopenia during CSFV infections [19,33]. Existing in vivo cytokine studies on CSFV mainly focus on highly virulent CSFV strains and/or vaccination trials [40,42]. However, von Rosen et al. [43] used lowly, moderately, and highly virulent CSFV strains in their cytokine study. The TNF-α response of the infected pigs in our study generally correlated to that observed by von Rosen et al. [43], who inoculated with CSFV strains Lithuania and Bergen with low and moderate virulence, respectively. The TNF-α concentrations detected in our study were generally higher than seen in pigs with mild clinical disease after infection with the Eystrup strain [4] (J. Nielsen, unpublished results) but considerably lower than after infection with the highly virulent Koslov strain. Thus, in Koslov-infected pigs, Renson et al. [42] detected TNF-α values ranging between 345 and 1703 pg/mL, and in a previous study, we detected peak mean TNF-α values up to 1347 pg/mL, with variations for individual pigs reaching up to 1031, 1590, and 4143 pg/mL in severely diseased pigs on PID 7 immediately before euthanasia due to welfare reasons (J. Nielsen, unpublished results). The changes in TNF-α observed in the various studies strongly indicate that the maximum TNF-α levels correlate with the disease outcome, i.e., high levels of TNF-α occur with severe disease. This was also supported by the body temperatures and CS since temperature elevation coincided with increased CS, CRP, and TNF-α levels. To our knowledge, this is the first report of CRP and TNF-α inflammatory responses during experimental infection with the CSFV strain Paderborn. Together with the demonstrated changes in the various leukocyte sub-populations, our study contributes to the elucidation of the immunopathological mechanisms active during CSFV infections. 

## 5. Conclusions

In conclusion, we saw a lack of consistency in several biological parameters used for analyzing the progression and outcome of CSFV infection when comparing oral and intranasal inoculation in an experimental setup using the moderately virulent CSFV strain Paderborn. Intranasal application provided more robust and reproducible results than oral application. This implies that it is essential to consider experimental design and, in particular, the route of inoculation for comparative CSFV studies.

## Figures and Tables

**Figure 1 vetsci-11-00056-f001:**
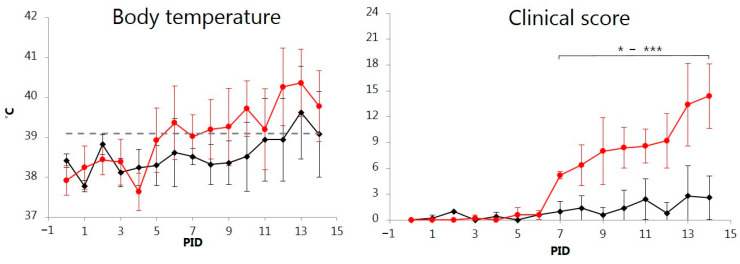
Development of body temperature (measured by subcutaneous microchip transponder) and clinical scores for pigs orally (ORAL group; ◆) or intranasally (NASAL group; ●) inoculated with the classical swine fever virus strain Paderborn. Each point represents the mean ± SD of 5 piglets. Temperatures above the grey dashed horizontal line at 39.1 °C correspond to fever. PID = post-inoculation day. Significant different (*p* < 0.05 to *p* < 0.001) means between groups are marked by * to ***.

**Figure 2 vetsci-11-00056-f002:**
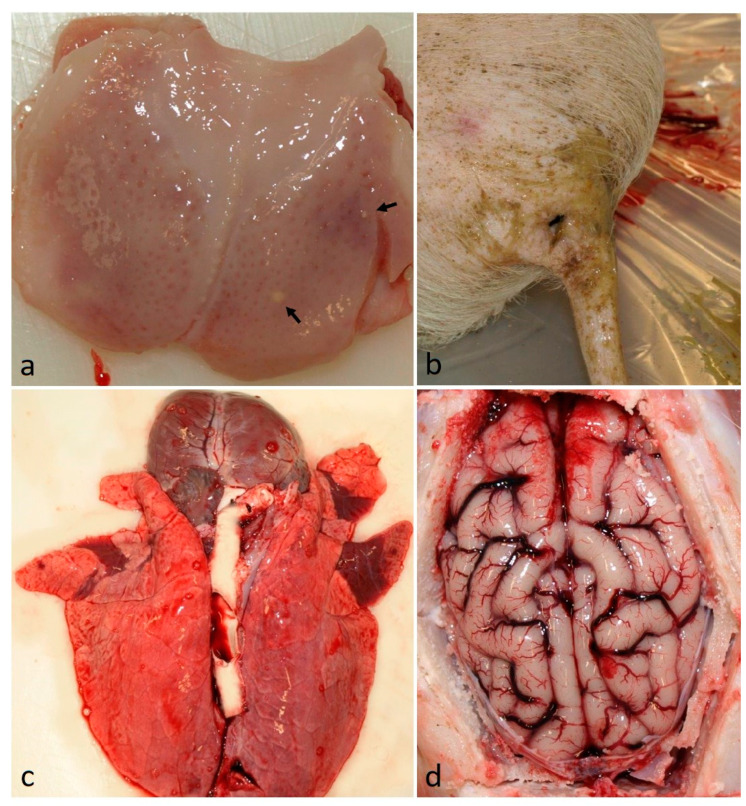
Gross pathology pictures of pig No. 8 inoculated intranasally with classical swine fever strain Paderborn. (**a**) Tonsil with bluish discoloration, erythema, and pustules (arrows). (**b**) Tail and hindquarter soiled with watery, greenish feces consistent with diarrhea. (**c**) Lung with lobular bronchopneumonia in the cranioventral lung lobes. (**d**) Brain with meningeal hyperemia and hemorrhage.

**Figure 3 vetsci-11-00056-f003:**
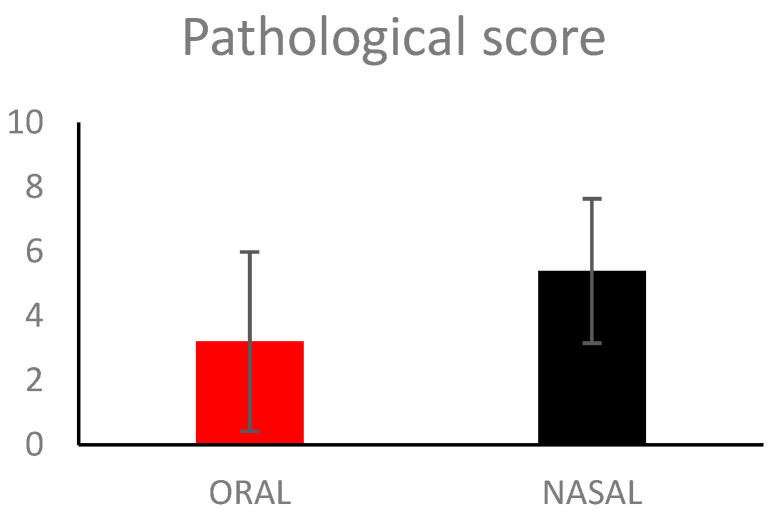
Pathological score. Mean values and standard deviation of five pigs inoculated orally (ORAL) or intranasally (NASAL) with classical swine fever strain Paderborn.

**Figure 4 vetsci-11-00056-f004:**
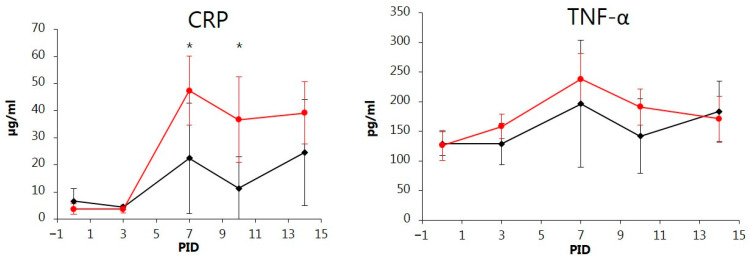
Variations over time in the concentrations of C-reactive protein (CRP) and TNF-α in peripheral blood of pigs orally (ORAL group; ◆) or intranasally (NASAL group; ●) inoculated with the classical swine fever virus strain Paderborn. Each point represents the mean ± SD of 5 piglets. PID = post-inoculation day. Significant different (*p* < 0.05) means between groups are marked by *.

**Figure 5 vetsci-11-00056-f005:**
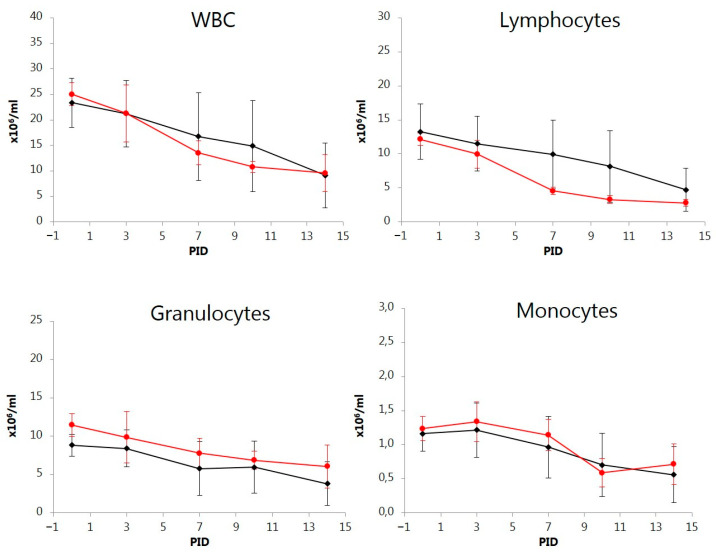
Variations over time in absolute numbers of total white blood cells (WBCs), lymphocytes, granulocytes, and monocytes of pigs orally (ORAL group; ◆) or intranasally (NASAL group; ●) inoculated with the classical swine fever virus strain Paderborn. Each point represents the mean ± SD of 5 piglets. PID = post-inoculation day.

**Figure 6 vetsci-11-00056-f006:**
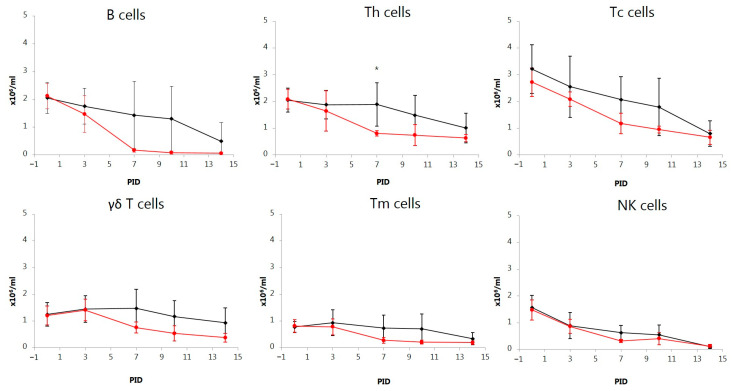
Variations over time in absolute numbers of B cells and T cell sub-populations (T helper (Th) cells; cytotoxic T (Tc) cells; γδ T cells; memory Th (Tm) cells; natural killer (NK) cells) of pigs orally (ORAL group; ◆) or intranasally (NASAL group; ●) inoculated with the classical swine fever virus strain Paderborn. Each point represents the mean ± SD of 5 piglets. PID = post-inoculation day. Significant different (*p* < 0.05) means between groups are marked by *.

**Table 1 vetsci-11-00056-t001:** Virological results reported as PCR Ct values in serum/virus isolation log_10_ titer TCID_50_ per 50 µL serum/virus isolation log_10_ titer TCID_50_ per 50 µL tonsillar scraping from pigs orally or intranasally inoculated with the classical swine fever virus strain Paderborn.

	PID	Pig No.				
Oral inoculation		1	2	3	4	5
	0	-/-/nd	-/-/nd	-/-/nd	-/-/nd	-/-/nd
	3	-/-/-	39/-/-	-/-/-	-/-/-	-/-/-
	7	36/1.5/-	37/2.7/1.5	-/-/-	-/-/-	-/-/-
	10	30/3.5/-	32/4.2/1.7	-/-/-	-/-/-	-/-/-
	14	27/2.5/0.8	27/3.2/1.5	-/-/-	-/-/-	38/1.3/-
Intranasal inoculation		6	7	8	9	10
	0	-/-/nd	-/-/nd	-/-/nd	-/-/nd	-/-/nd
	3	-/-/-	-/-/-	-/-/-	-/-/-	-/-/-
	7	36/2.7/2.2	36/2.7/1.5	32/3.7/1.8	36/3.2/-	36/2.7/-
	10	30/4.5/1.5	31/5.0/1.7	26/4.5/2.7	30/4.2/1.7	30/3.5/1.5
	14	27/4.7/3.2	27/5.7/3.2	26/4.5/1.7	30/3.0/1.5	26/3.5/2.7

PID = post-inoculation day; - = negative result; nd = not done.

## Data Availability

The datasets used for this study are available from the corresponding author upon reasonable request.

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
