# Peer review of "Intranasal Inoculation with Classical Swine Fever Virus Provided a More Consistent Experimental Disease Model Compared to Oral Inoculation"

_vetsci, 2024, doi:10.3390/vetsci11020056_

Round 1

Reviewer 1 Report

Comments and Suggestions for Authors

The different inoculation route of classic swine fever viruses related to different outcomes. This paper compared the outcomes in young pigs between oral and intranasal inoculation. More pronounced clinical symptoms, pathological lesions, haematological and immunological parameters were observed in the intranasal infected pigs. The results indicated that intranasal inoculation route was more suitable for pathogenicity study.

But there are still several problems that need to be improved:

1. This study compared two inoculation route of classic swine fever virus, by intranasal and by oral inoculation. But there’s still another inoculation method, intramuscularly injected, commonly used in classic swine fever virus pathogenicity study. It is better to add comparing this method with intranasal and oral inoculation in the “introduction” or “discussion” part.

2.The writing of this article should be carefully modified, some parts were difficult to understand.  

3.  In line 116 “Rectal body temperatures were measured on PID 1 (NASAL group), 3 (ORAL group)”. Why were the temperatures measured differently between the two groups?

4.  In line 109 “the experiment was ended at PID 14”, but in line 173 “a total of 36 cumulative days with fever was recorded for the pigs in the NASAL group” How could there be 36 days?

Reviewer 2 Report

Comments and Suggestions for Authors

CSFV is a major health threat for pig population globally. Here the authors showed that when inoculated by intranasal route, the virus  caused more infection in the experimental disease model compared to oral route of inoculation. Its a well-designed study with robust analysis and methodology, and documentation.

Comments:

The major point missing in this study is why inoculation by the intranasal route caused more infection compared to the oral route?Maybe there are more receptors of this virus n cells of intranasal route compared to the oral route? what is the speculation?

Other comments:

Pig number, do you think using 5 pig in each group is enough?Although SPF pigs were used, bt why not kept tow or more pig without inoculation as true control?

What is the rational of using 10^5 TCID50 dose for infection?

Why only a moderately virulent CSFV strain was used, not any virulent, may be the findings will not be same if a virulent virus strain was used?

Please add few figure of necropsy findings, both control and affected side by side.

In most of the Figures, the SD/SE is too much, any explanations?

Please add SE/SD in the graphs in the Suppl. doc

Reviewer 3 Report

Comments and Suggestions for Authors

The severity of classical swine fever virus infections in pigs is determined primarily by the virulence of the virus but is also dependent on host factors and other environmental factors including the experimental procedures. With the aim of tackling factors influencing the reproducibility of experimental CSFV infections, the authors compared the effect of oral versus intranasal inoculation on disease outcome using the moderately virulent Paderborn strain. For this, they infected two groups of 5 pigs via the oral or nasal route, respectively. They monitored clinical parameters, gross pathology, virus in tonsil scrapings and serum, C-reactive protein, TNF, seroconversion and finally hematological and immunological parameters. They show convincingly that intranasal inoculation provides more robust and reproducible results than oral inoculation. They provide also a comprehensive description of various serological, hematological and immunological parameters related to moderate CSF.

General assessment:

Several routes of CSFV infection are described in the literature, but to the best of my knowledge, this is the first study that compares the outcome of oral and nasal inoculation side by side. This is valuable information that may guide future studies towards using nasal inoculation for pathogenesis studies. It has also an impact on assessing bait vaccination via to oral route (as discussed on lines 338-348). Some studies used intramuscular CSFV application for reproducible infection. However, natural inoculation such as nasal application is in any case preferable for pathogenesis studies. In that sense, this study is important and worth being published. For the rest, this study remains very descriptive and lacks novelty overall. The authors measured a lot of different parameters, for which only clinical and virological data and CRP as well were significantly different between the two groups to support the main conclusion of the paper. The pathological findings were not compared statistically since no pathological score was used, although more severe lesions were noted subjectively with nasally infected pigs. The other parameters behave as expected from the numerous studies published with moderately virulent CSFV in pigs, but they do mostly not differ in relation to the two different inoculation routes. Nevertheless, the data were assessed correctly (except for some points to be clarified, see comments below). For a moderately virulent strain such as Paderborn, this is one of the most comprehensive descriptive analysis of disease parameters, which to my opinion justifies publication in Veterinary Sciences.

Major comments:

Comment 1: The statistical analyses should be described more precisely. The authors compare either the values between the two groups or the values between time 0 (not always specified) and later time points within the same group. Comparison between groups uses unpaired t-tests while significance of the decrease of mean values within the same group over time is assessed typically with paired t-tests. This is not mentioned anywhere and should be described properly in section 2.7. 

Comment 2: Related to the previous comment, in sections 3.6. and 3.7. it is not always clear what is compared with what. For instance, on lines 269-273 (like elsewhere in these two sections), the significant decrease at the specified times refers implicitly to time 0, but this is not specified. Please clarify.

Comment 3: In figure 1 (body temperature and clinical score) and figure 2 (CRP), statistical comparison between the two groups was probably done for each time point. It would be helpful to mark with stars or even better to display the precise p value for each time at which the difference between the groups was significant. An alternative would be to compare statistically the area under the curve.

Comment 4: it is a pity that the authors did not apply a pathological score to assess the lesions. Is there a possibility to determine the pathological score retrospectively with the data available? This would allow statistical evaluation of the oral versus nasal group.

Minor comments:

Comment 5: line 20, write “carried out for decades”

Comment 6: “respectively” should be deleted on line 27. Otherwise, it sounds as clinical disease parameters were measured in orally infected pigs and indicators of disease were assessed in nasally infected pigs, which doesn’t make sense of course.

Comment 7: on line 30, “of importance for combat of the disease” is grammatically wrong. There are numerous mistakes like this throughout the text. The English must be improved.

Comment 8: the authors should specify whether the 10 pigs divided randomly into two groups of 5 pigs are from one or more litters and provide information of sex distribution. Did they block for even distribution of males (probably castrates) and females within the two groups?

Comment 9: the inoculation of the two groups with the same virus is not described properly (lines 95-100). I assume that the 10 pigs were inoculated from a single pool of virus (5 by oral and 5 by nasal application) that was then back-titrated, which is crucial for the comparison to be valid. The way it is described does not specify this clearly. It says that the ORAL group was inoculated with 4ml of virus per animal and the virus was back-titrated. Next sentence says that the NASAL group was inoculated with “equivalent doses of virus”, which does not explicitly imply that they were infected from the same pool. Please clarify.

Comment 10: on lines 353 and 404, the CSFV strain is Koslov (not Kozlov)

Comments on the Quality of English Language

Language editing is required
